# Feasibility, acceptability, and efficacy of a positive emotion regulation intervention to promote resilience for healthcare workers during the COVID-19 pandemic: A randomized controlled trial

**Judith Tedlie Moskowitz**[1]\*, **Kathryn L. Jackson**[1], **Peter Cummings**[1], **Elizabeth L. Addington**[1], **Melanie E. Freedman**[1], **Jacquelyn Bannon**[2], **Cerina Lee**[2], **Thanh Huyen Vu**[3], **Amisha Wallia**[4,5], **Lisa R. Hirschhorn**[1], **John T. Wilkins**[3,6], **Charlesnika Evans**[3,7]

1 Department of Medical Social Sciences, Feinberg School of Medicine, Northwestern University, Chicago, IL, United States of America, 2 Institute for Public Health and Medicine, Center for Education in Health Sciences, Northwestern University, Chicago, IL, United States of America, 3 Department of Preventive Medicine, Feinberg School of Medicine, Northwestern University, Chicago, IL, United States of America, 4 Center for Health Services and Outcomes Research, Feinberg School of Medicine, Institute for Public Health and Medicine, Northwestern University, Chicago, IL, United States of America, 5 Department of Medicine, Division of Endocrinology, Metabolism, and Molecular Medicine, Feinberg School of Medicine, Northwestern University, Chicago, IL, United States of America, 6 Department of Medicine, Division of Cardiology, Feinberg School of Medicine, Northwestern University, Chicago, IL, United States of America, 7 Department of Veterans Affairs, Center of Innovation for Complex Chronic Healthcare, Edward Hines, Jr. VA Hospital, Hines, IL, United States of America

\* judith.moskowitz@northwestern.edu

## Abstract

### Introduction

Burnout poses a substantial, ongoing threat to healthcare worker (HCW) wellbeing and to the delivery of safe, quality healthcare. While systemic and organization-level changes in healthcare are critically important, HCWs also need individual-level skills to promote resilience. The objective of this trial is to test feasibility, acceptability, and efficacy of PARK, an online self-guided positive affect regulation intervention, in a sample of healthcare workers during the COVID-19 pandemic.

### Design and methods

In the context of the unprecedented rise in burnout during the COVID-19 pandemic, we conducted a randomized waitlist-controlled trial of the Positive Affect Regulation sKills (PARK) program—a five-week, online, self-guided coping skills intervention nested within an ongoing cohort of HCWs. N = 554 healthcare workers were randomly assigned to receive the intervention immediately or to receive the intervention after approximately 12 weeks. Outcomes included change in burnout, emotional wellbeing (positive affect, meaning and purpose, depression, anxiety) and sleep over approximately 12 weeks. Analyses included

**Data Availability Statement:** We have uploaded the de-identified dataset to OSF: https://osf.io/bvdu5/that.

**Funding:** Peter G. Peterson Foundation #21048 Funding Acquisition: CE, JTW, LRH, AW, JTM https://www.northwestern.edu/petersonfund/ The funders had no role in study design, data collection and analysis, decision to publish, or preparation of the manuscript.

**Competing interests:** The authors have declared that no competing interests exist.

mixed-effects linear regression models comparing change over time in outcomes between intervention and control conditions.

## Results

One third (n = 554) of the participants in the cohort of HCWs consented to participate and enrolled in PARK in April 2022. Compared to those who did not enroll, participants in the trial reported higher burnout, poorer emotional wellbeing, and poorer sleep at baseline (April, 2022; all $p$s < .05). Intent-to-treat analyses showed that participants randomly assigned to the intervention immediately (PARK-Now) improved significantly on anxiety (within-group change on PROMIS T-score = -0.63; $p$ = .003) whereas those in the waitlist (PARK-Later) did not (within group T-score change 0.04, $p$ = 0.90). The between-group difference in change, however, was not statistically significant ($B$ = -0.67 $p$ = 0.10). None of the other well-being outcomes changed significantly in the intervention group compared to the waitlist. Additional as-treated analyses indicated that those participants who completed all 5 of the weekly online lessons (N = 52; 9.4%) improved significantly more on the primary outcome of positive affect compared to those who enrolled in PARK but completed zero lessons (n = 237; 42.8%; B = 2.85; p = .0001).

## Conclusions

Online self-guided coping skills interventions like PARK can be effective in targeted samples and future work will focus on adaptations to increase engagement and tailor PARK for HCWs who could most benefit.

## Background and objectives

Healthcare workers' (HCW) well-being is a national priority [1]. Multiple organizations and healthcare leaders are sounding alarms about rates of burnout and mental health concerns in these critically important workforces [2–5]. Burnout includes emotional exhaustion and disengagement from work [6–8] and most studies reported HCW burnout rates ranging between 30–60% prior to the COVID-19 pandemic [9–12]. Recently, burnout in HCWs has increased, with rates as high as 79% in some studies [13–16]. Burnout poses a substantial, ongoing threat to HCW well-being and to the delivery of safe, quality healthcare [17–21]. It is strongly correlated with both depression and anxiety [22, 23], as well as a host of negative physical health outcomes, including poor sleep, risky health behaviors (e.g. reduced physical activity, increased substance use), and physical health concerns such as headaches and GI disturbances [12, 24–29]. Further, HCW burnout and low levels of well-being are associated with reduced effectiveness in healthcare delivery and increased risk to patient wellbeing. Meta-analyses show links between burnout and reduced quality and safety of care, including higher likelihood of medical errors [30, 31].

Research prior to the COVID-19 pandemic showed greater resilience and lower levels of burnout and depressive symptoms among HCWs with higher levels of positive psychological factors such as positive affect and meaning and purpose [32–34]. Positive affect, defined as subjective positive emotional states such as joy, contentment, and happiness, is associated with higher job satisfaction and may buffer work-related stress and reduce burnout [33, 35, 36]. Research conducted during the pandemic similarly showed that positive affect can mitigate

against negative effects of stress on HCW wellbeing outcomes such as depression [37, 38]. Evidence-based interventions for improving and maintaining HCW well-being are necessary [17] and while systemic and organization-level changes in healthcare are critically important, HCWs also need individual-level skills to promote resilience [3]. Accordingly, the US Surgeon General has prioritized the promotion of HCWs' positive psychological wellbeing as an essential pathway to reduce burnout and ensure quality healthcare [17].

Interventions that specifically target positive affect help maintain well-being in the midst of stress [39–42] and may help prevent HCW burnout [37] and thereby reduce deleterious health effects among HCWs. The Positive Affect Regulation sKills (PARK) program [39] is an evidence-based intervention, delivered via a self-guided, web-based platform, in which participants learn empirically-supported skills for increasing daily experiences of positive affect. The purpose of the present study was to test feasibility, acceptability, and efficacy of PARK in a sample of healthcare workers during the COVID-19 pandemic. Guided by the Positive Pathways to Health theoretical model [43], we hypothesized that HCWs who were randomized to receive the PARK intervention would report greater increases in positive affect compared to a waitlist control group. Furthermore, we hypothesized that PARK participants would report decreased burnout, depression, and anxiety as well as increased meaning and purpose, and improved sleep compared to the waitlist. We explored intervention engagement (number of weeks completed), baseline levels of wellbeing, and sociodemographics as moderators of the intervention effect on our primary outcome of positive affect.

## Methods

PARK HCW was a two-group, randomized, waitlist-controlled trial nested within the Northwestern Medicine Healthcare Worker SARS-CoV-2 Serology Study, an ongoing, prospective cohort study of healthcare workers from a large, tertiary academic medical center during the COVID-19 pandemic. The study is described in detail elsewhere [44, 45]. Briefly, the cohort is comprised of HCWs from 10 hospitals, 18 immediate care centers, and 325 outpatient practices in the Chicago area and surrounding suburbs. At cohort inception (May 2020), 6,510 HCWs enrolled in the study. The initial objective of the cohort study was to perform surveillance for SARS-CoV-2 infection and determine work and community characteristics that predicted SARS-CoV-2 in HCWs. In the second year of the cohort (June 2021), 3,568 participants re-consented to enroll for continued follow-up. At that point, we added measures to assess psychological wellbeing and job-related burnout [37]. Participants in this reconsented sample of the HCW cohort were invited to participate in the present trial and completed surveys in April 2022 (Baseline/T1 completed between April 1, 2022 and April 30, 2022), July 2022 (T2), and October 2022 (T3).

Human subjects approval was obtained from Northwestern University's IRB (STU00212515-MOD0025) and the study was pre-registered at clinicaltrials.gov (NCT05394051). All participants completed online informed consent for the PARK substudy as well as for the parent study. We used the CONSORT checklist when writing our report [46].

### Procedures

In April 2022, participants in the cohort who reconsented (N = 3,568) were sent an email with a video introducing the PARK intervention, inviting them to participate in the randomized controlled trial nested within the ongoing cohort surveys. The email invitation included a link to the PARK study's REDCap informed consent landing page. Participants were randomized 1:1 to PARK-Now, the group that received the PARK intervention immediately, or to PARK-Later, a waitlist control condition who received access to PARK approximately 3 months later.

**Table 1. Overview of PARK intervention.**

| Week | Skills | Goals | Practice Exercises |
|---|---|---|---|
| 1 | **Positive events, Savoring, and Gratitude** | Recognize positive events and associated positive emotions; practice amplifying the experience of positive events; learn to practice gratitude | Note a positive event each day and write about it (savoring); start a daily gratitude journal |
| 2 | **Mindfulness** | Learn and practice the awareness and nonjudgment components of mindfulness | Daily informal mindfulness activities and 10-minute formal breath awareness activity |
| 3 | **Positive Reappraisal** | Understand positive reappraisal & how it can lead to increased positive emotions in the face of stress | Report a relatively minor stressor each day, then list ways it can be positively reappraised |
| 4 | **Personal Strengths, Attainable Goals** | Recognize personal strengths, skills, or talents; Understand benefits of goals that are appropriately challenging but still feasible | List a strength each day and how it was "expressed" behaviorally; work toward an attainable goal and note progress each day |
| 5 | **Self-Compassion** | Learn benefits of being kind and understanding toward oneself, even when under stress | Practice changing harsh self-criticism to self-compassion |

The randomization sequence was generated by the statistician based on a computer algorithm. Participants who were randomized to PARK-Now received an email within one week of consenting with instructions to create an account and access the PARK skills on the online platform. PARK-Later (waitlist) participants received access to the PARK skills following the July 2022 cohort survey. HCW participants who did not consent to participate in PARK were considered an additional comparison condition.

**PARK intervention.** The PARK intervention is a self-guided, online program that is delivered over 5 weeks and covers 8 skills that specifically target positive affect (see Table 1). Each week includes one or two brief (10–15 minute) didactic sessions with exercises for practicing the skills. Participants could not skip ahead, but they could return to old lessons or exercises if they chose to do so. During the 5 weeks of skill delivery, participants were sent daily email reminders to complete the skills practice activities. PARK has been shown to be efficacious in a general population sample during COVID [39] and was not tailored specifically for HCW.

## Measures

Participants self-reported demographics (age, sex and race/ethnicity) and occupation (i.e. physician, nurse, administration etc.). *Burnout* was measured using the Oldenburg Burnout Inventory (OLBI). The OLBI is a well-validated and reliable 16-item measure which assesses two dimensions of burnout: exhaustion and disengagement from work [6, 7]. Item responses range from 1 (totally disagree) to 4 (totally agree), with statements such as, "During my work, I often feel emotionally drained". Total scores range from 16–64; scores were kept continuous for the present analyses. Other wellbeing measures of positive affect, meaning and purpose, anxiety, depression, and social isolation were measured using Patient-Reported Outcomes Measurement Information System (PROMIS) [47–49] computer adaptive tests (CATs). All of these PROMIS measures utilize T-scores (Mean = 50, SD = 10) relative to the general population. PROMIS cut points (e.g., within normal limits, mild, moderate, severe) allow for interpretation of clinical significance of T-scores; a within-group change or between-group difference of 3 T-score points is accepted to represent a meaningful change or difference [50]. Sleep was measured using the ASCQ-Me Sleep Impact CAT. ASCQ-Me is an Item Response Theory-based measurement system originally designed for use in populations with sickle cell disease but has been shown to perform similarly in the general population [51]. As with PROMIS, this measure is also scored on a T-score metric (Mean = 50; SD = 10); with higher scores indicating better sleep.

**Feasibility and acceptability.** Platform usage metrics were used as indicators of feasibility. Specifically, usage was calculated based on number of weeks the skills were completed (0–5). We categorized usage into none (0 weeks complete) some (1–4 weeks), and all (5 weeks). To assess acceptability, participants were asked reasons for not completing all the skills lessons and, separately, reasons for not completing all the home practice. For both questions, participants were provided a set of possible reasons and an option to write in additional reasons for not completing the skills. In addition, participants were asked to rate PARK on a series of questions [52] to assess acceptability and relative advantage utilizing a 5-point Likert scale (completely agree, agree, neither agree nor disagree, disagree, completely disagree). Those questions were: *1) I liked participating in PARK, 2) PARK is easy to understand and use, 3) I welcome PARK as an intervention to improve wellness, 4) Healthcare workers would really benefit from PARK, 5) PARK is more effective than interventions currently available to support wellness in HCWs*, and 6) *PARK would be successful in reducing burnout for HCWs in my institution*. Finally, we asked participants "What suggestions do you have to improve the PARK program in terms of content or delivery?"

## Statistical analysis

Descriptive statistics at baseline were calculated for the overall sample, and separately for PARK-Now and PARK-Later intervention groups, and by usage category (no, some, all). Differences in baseline characteristics were assessed using chi-squared test for categorical and t-tests for continuous variables.

**Intent to treat analyses.** The effect of the PARK intervention on change in psychological wellbeing (positive affect, meaning and purpose, anxiety, depression, social isolation), burnout, and sleep was assessed using independent mixed-effects linear regression models, including fixed effects for group (PARK-Now vs PARK-Later), time (T1/April, 2022, T2/July,2022), and the interaction of group*time; a significant group*time interaction indicated significant difference in change over time between groups. Random intercepts were included to allow participant baseline scores to vary from the average. Least-squares mean (LSM) estimates of the outcome at each time point and for the change across time were calculated for each intervention group; Cohen's d [53] effect sizes were calculated for the difference in change across time between the two intervention groups. Due to significant dropout at T2, missing outcome data at T2 was replaced using the single imputation method Last Observation Carried Forward (LOCF) [54]. LOCF was used to impute missing data, as we expected that participants in good health and with improving health were likely to be the ones who dropped out. In addition, we used a mixed effects modeling approach and compared analysis results with and without LOCF as a sensitivity analysis, as recommended by the panel on handling missing data in clinical trials [54]. Results remained similar in magnitude and significance across both methods, and thus, we considered the LOCF approach justified.

**As treated analyses.** To examine whether usage (number of weeks of lessons completed) was related to intervention effects, a series of "as treated" analyses were completed on a combined sample in which PARK-Now and PARK-Later groups were combined and aligned such that T1 for PARK-Now and T2 for PARK-Later became "pre-intervention," and T2 for PARK-Now and T3 for PARK-Later became "post-intervention." Usage of PARK was categorized into no, some, or all weeks completed as described above. To assess the effect of PARK usage on change in psychological outcomes, independent mixed effect linear regression models including fixed effects for usage group, time, and usage group*time and random intercept were run on the combined sample. Additionally, to account for potential variance due to temporal differences in the PARK-Now and PARK-Later sample including COVID-19 pandemic levels, a variable for original

assignment was included as a covariate. LSM estimates of the outcome at each time point and change in outcome were calculated and compared between usage groups; Cohen's d [53] effect sizes were calculated for the difference in change across time between usage groups. As with the ITT analyses, LOCF was used to impute missing post-intervention outcome data.

**Exploratory analyses.** We explored age, gender, HCW role, baseline burnout and other indicators of wellbeing (meaning and purpose, depression, anxiety, and social isolation) as potential moderators of the effect of PARK on change in the primary outcome (positive affect) over time. We used a series of independent mixed effects regression models run on the combined PARK-Now and PARK-Later sample. These models included fixed effects of time, moderator, and moderator*time, as well as a random intercept and covariate for original intervention assignment. A significant moderator*time effect indicated significant difference in change in positive affect over time by level of the moderator. Significant moderators (along with relevant interactions) were then independently added into the original ITT analysis model to test the moderation effect of intervention on change in positive affect. In these models, a significant three-way interaction (moderator*group*time) indicated significant moderation in the effect of intervention on change in positive affect over time. T-scored moderators were converted to theta scores (theta = (t-score-50)/10) to provide a meaningful "zero" for effects. LSM estimates for the change in positive affect over time for intervention and control groups were calculated at low (-1 SD), average (mean = 0), and high (+1 SD) levels of the moderator for each model.

All analyses were completed using SAS v. 9.4. P-values of <0.05 were considered statistically significant in all analyses.

## Results

Of the 3,568 participants who were sent the cohort survey in April 2022, 1,685 (47%) responded to the invitation to the PARK intervention. Of those who responded to the invitation, 554 (33%) agreed to participate. The CONSORT diagram [55] is in Fig 1 and illustrates the number of participants who completed the survey at each assessment point, and the proportion of participants who completed none, 1–4, or all of the intervention sessions (see Fig 1, CONSORT diagram). PARK participants were randomized to have access to PARK immediately (PARK-Now) or to the waitlist control (PARK-Later) where they received access to the PARK platform 3 months later (after the next cohort survey wave).

Baseline comparisons of those who did and did not enroll in PARK, as well as between those randomized to PARK-Now vs PARK-Later are shown in Table 2. At Baseline/T1, those who enrolled in PARK reported significantly higher anxiety, depression, social isolation, and burnout; significantly lower positive affect and meaning and purpose; and poorer sleep compared to those who did not enroll in PARK. In addition, those who enrolled in PARK were younger and more likely to be female.

Among those who enrolled in PARK, those who were randomized to PARK-Now did not differ from the waitlist (PARK-Later) on any of the wellbeing, burnout, or sleep impact measures. PARK-Now participants only differed from PARK-Later participants in that, on average, they were older (*p* = 0.02).

### Intent-to-treat analyses

Results of the Intent-to-Treat analyses are shown in Table 3. Within the PARK-Now group, positive affect and anxiety both improved significantly from baseline/T1 (April 2022) to the post/T2 (July 2022) assessment (p = 0.006, 0.031, respectively). However, neither of these improvements were statistically significantly greater than the waitlist (PARK-Later) group over the same period (group*time interaction p = 0.60, 0.10,

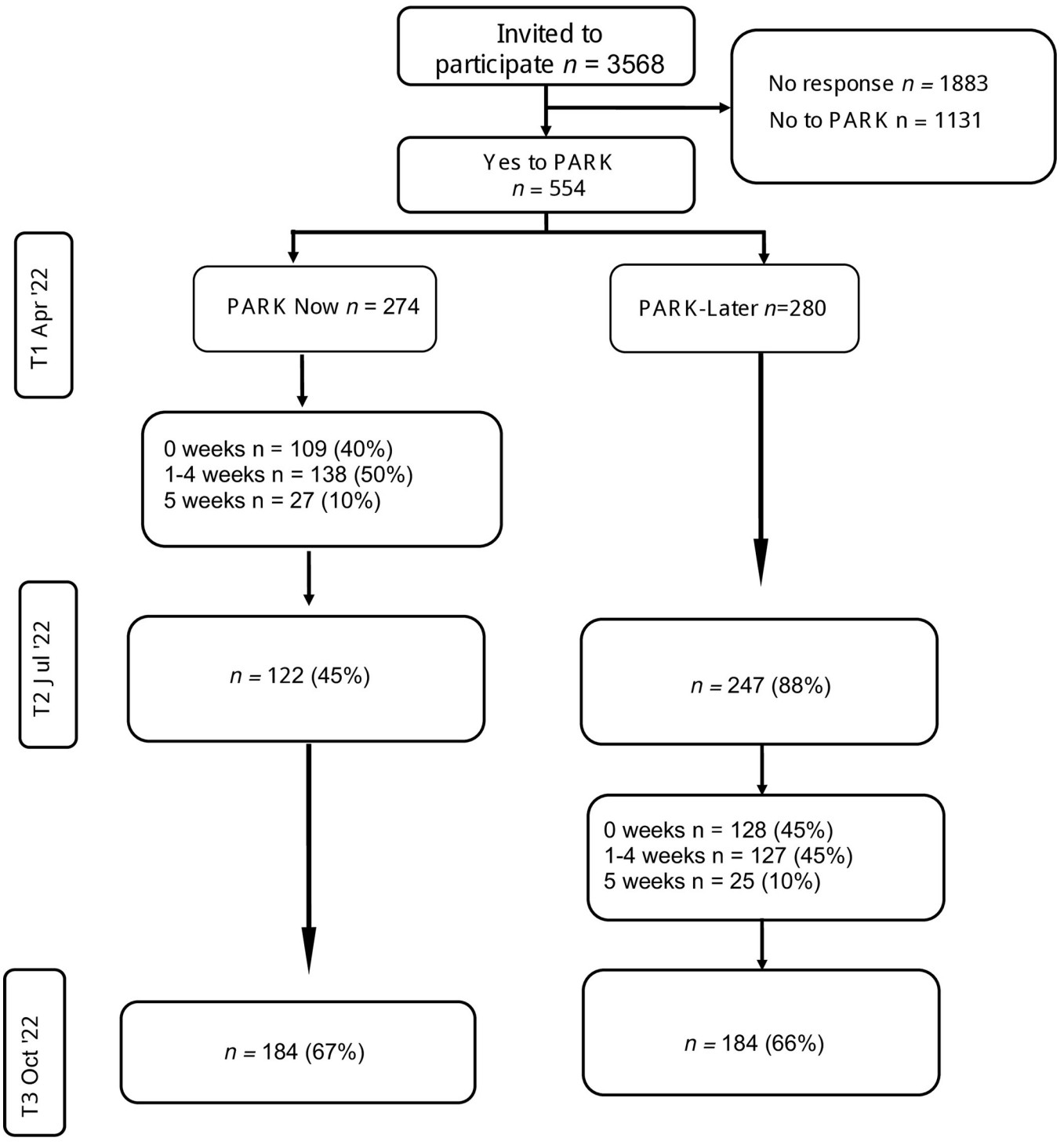

**Fig 1. PARK healthcare worker CONSORT.**

respectively). Specifically, for positive affect, both the PARK-Now and PARK-Later groups improved significantly and did not differ from each other. See Fig 2 for group means and 95% confidence intervals over time. Note that scores for cohort participants who declined PARK are included in the figure but were not part of the statistical analysis. While both

**Table 2. Baseline/T1 (April 2022) comparisons for agreeing to enroll in PARK (No vs Yes), N = 1685 and randomization (Now vs Later), N = 554.**

| | Baseline/T1 (April 2022) (N = 1685) | | | PARK randomization (N = 554) | | |
|---|---|---|---|---|---|---|
| | **PARK NO (N = 1131)** | **PARK YES (N = 554)** | **p** | **Now (N = 274)** | **Later (N = 280)** | **p** |
| Socio-demographics* | | | | | | |
| Age | 45.11 (11.93) | 43.51 (10.59) | 0.01 | 44.58 (10.34) | 42.46 (10.74) | 0.02 |
| Sex | | | | | | |
| Male | 216 (19.22%) | 72 (13.09%) | | 29 (10.70%) | 43 (15.41%) | |
| Female | 908 (80.78%) | 478 (86.91%) | <0.01 | 242 (89.30%) | 236 (84.59%) | 0.10 |
| Race | | | | | | |
| White | 967 (87.20%) | 472 (88.06%) | | 231 (87.50%) | 241 (88.60%) | |
| Black | 27 (2.43%) | 17 (3.17%) | | 9 (3.41%) | 8 (2.94%) | |
| Asian | 111 (10.01%) | 42 (7.84%) | | 21 (7.95%) | 21 (7.72%) | |
| Other | 4 (0.36%) | 5 (0.93%) | 0.20 | 3 (1.14%) | 2 (0.74%) | 0.95 |
| Wellbeing | | | | | | |
| Anxiety | 51.59 (8.00) | 55.40 (7.50) | <0.0001 | 55.11 (7.15) | 55.68 (7.82) | 0.37 |
| Depression | 47.86 (7.51) | 50.90 (7.27) | <0.0001 | 50.55 (7.39) | 51.24 (7.14) | 0.26 |
| Social Isolation | 41.86 (8.71) | 46.07 (9.18) | <0.0001 | 45.52 (9.17) | 46.60 (9.17) | 0.17 |
| Positive Affect | 48.84 (9.17) | 45.88 (8.68) | <0.0001 | 45.96 (8.93) | 45.80 (8.44) | 0.83 |
| Meaning & Purpose | 54.85 (10.38) | 51.39 (10.31) | <0.0001 | 51.70 (10.87) | 51.07 (9.73) | 0.47 |
| Burnout (Total) | 35.08 (7.63) | 37.44 (7.74) | <0.0001 | 36.82 (7.45) | 38.04 (7.97) | 0.06 |
| Sleep Impact | 59.02 (7.10) | 57.50 (7.39) | <0.0001 | 57.67 (7.22) | 57.32 (7.55) | 0.57 |
| Occupation | | | | | | |
| Administrative | 240 (21.30%) | 116 (21.13%) | | 63 (23.16%) | 53 (19.13%) | |
| Physician | 191 (16.95%) | 71 (12.93%) | | 31 (11.40%) | 40 (14.44%) | |
| APN, PA | 66 (5.86%) | 40 (7.29%) | | 23 (8.46%) | 17 (6.14%) | |
| RN | 276 (24.48%) | 152 (27.68%) | | 70 (25.73%) | 82 (29.60%) | |
| Allied Health | 250 (22.18% | 128 (23.32%) | | 67 (24.63%) | 61 (22.02%) | |
| Other | 104 (9.23%) | 42 (7.65%) | 0.17 | 18 (6.62%) | 24 (8.67%) | 0.42 |
| Patient Contact | | | | | | |
| No | 188 (19.18%) | 77 (15.81%) | | 38 (15.45%) | 39 (16.18%) | |
| Yes | 792 (80.82%) | 410 (84.19%) | 0.11 | 208 (84.55%) | 202 (83.82%) | 0.82 |

Notes: For all continuous measures, sample means are included with standard deviations in parentheses; for all categorical measures, sample sub-totals are included with percentages in parentheses. All p-values are included up to the tenths place or with 2 significant figures.

*All socio-demographic data was gathered at a pre-baseline data collection timepoint

Abbreviations: APN = Advanced practice nurse, PA = Physician's assistant, RN = Registered nurse or equivalent

PARK-Now and PARK-Later appear to decrease from T2 to T3, the within-group change is not statistically different from 0 and the between-group difference in decrease is not statistically significant.

Anxiety was stable from baseline (T1) to post (T2) in the PARK-Later group, but the decrease within the PARK-Now group did not differ significantly from the waitlist condition. Fig 3 shows group means and 95% confidence intervals over time. There was no significant change from T1 to T2 for any of the other outcomes measured.

## As-treated analyses

We examined engagement with the PARK platform by combining usage metrics for PARK-Now and PARK-Later groups as PARK-later participants were given access to the platform

**Table 3. Intent to treat analyses comparing PARK now to PARK later T1 (April, 2022) to T2 (July, 2022).**

| | | Apr-22 (Baseline) | | Jun-22 (Post 1) | | Change | | | P-value for Change within group | p-value for interaction | Cohen's d effect size |
|---|---|---|---|---|---|---|---|---|---|---|---|
| | | Mean | Std | Mean | Std | Mean | Std | 95% CI | | | |
| PROMIS—Positive Affect | Now | 45.96 | 0.53 | 46.97 | 0.53 | 1.02 | 0.36 | (0.30,1.73) | 0.006 | 0.60 | -0.04 |
| | Waitlist | 45.80 | 0.53 | 47.09 | 0.53 | 1.29 | 0.36 | (0.58,2.00) | 0.0004 | | |
| PROMIS—Meaning and Purpose | Now | 51.70 | 0.62 | 52.04 | 0.62 | 0.33 | 0.37 | (-0.39,1.06) | 0.366 | 0.46 | -0.06 |
| | Waitlist | 51.07 | 0.62 | 51.79 | 0.62 | 0.71 | 0.36 | (-0.002,1.43) | 0.051 | | |
| PROMIS—Depression | Now | 50.55 | 0.44 | 50.01 | 0.44 | -0.54 | 0.30 | (-1.12,0.05) | 0.072 | 0.14 | -0.13 |
| | Waitlist | 51.24 | 0.43 | 51.33 | 0.43 | 0.09 | 0.29 | (-0.50,0.67) | 0.765 | | |
| PROMIS—Anxiety | Now | 55.11 | 0.45 | 54.48 | 0.45 | -0.63 | 0.29 | (-1.21,-0.06) | 0.031 | 0.10 | -0.14 |
| | Waitlist | 55.68 | 0.45 | 55.72 | 0.45 | 0.04 | 0.29 | (-0.53,0.61) | 0.896 | | |
| PROMIS—Social Isolation | Now | 45.52 | 0.55 | 45.32 | 0.55 | -0.21 | 0.34 | (-0.88,0.47) | 0.550 | 0.89 | -0.01 |
| | Waitlist | 46.60 | 0.54 | 46.46 | 0.54 | -0.14 | 0.34 | (-0.81,0.53) | 0.678 | | |
| Burnout–Total Score | Now | 36.82 | 0.46 | 36.81 | 0.46 | -0.02 | 0.26 | (-0.53,0.49) | 0.942 | 0.87 | 0.01 |
| | Waitlist | 38.04 | 0.46 | 37.96 | 0.46 | -0.08 | 0.26 | (-0.58,0.42) | 0.763 | | |
| PROMIS—Sleep | Now | 57.67 | 0.44 | 58.17 | 0.44 | 0.50 | 0.28 | (-0.06,1.06) | 0.078 | 0.72 | 0.03 |
| | Waitlist | 57.32 | 0.44 | 57.68 | 0.44 | 0.36 | 0.28 | (-0.19,0.91) | 0.201 | | |

Note: Means provided are least-squares means estimated from the models

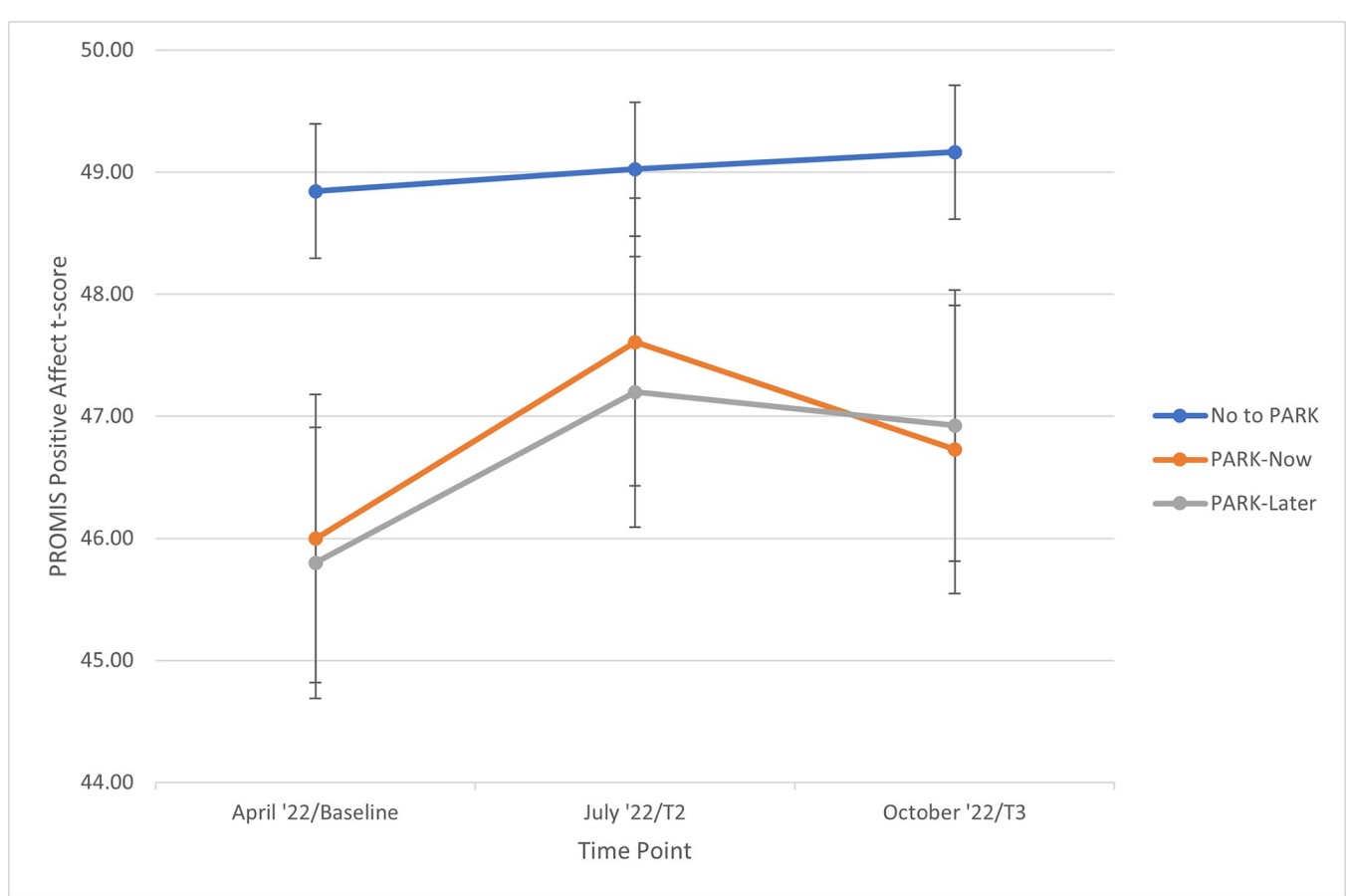

**Fig 2. Sample means with 95% CI for PROMIS positive affect t-scores by group at each timepoint.**

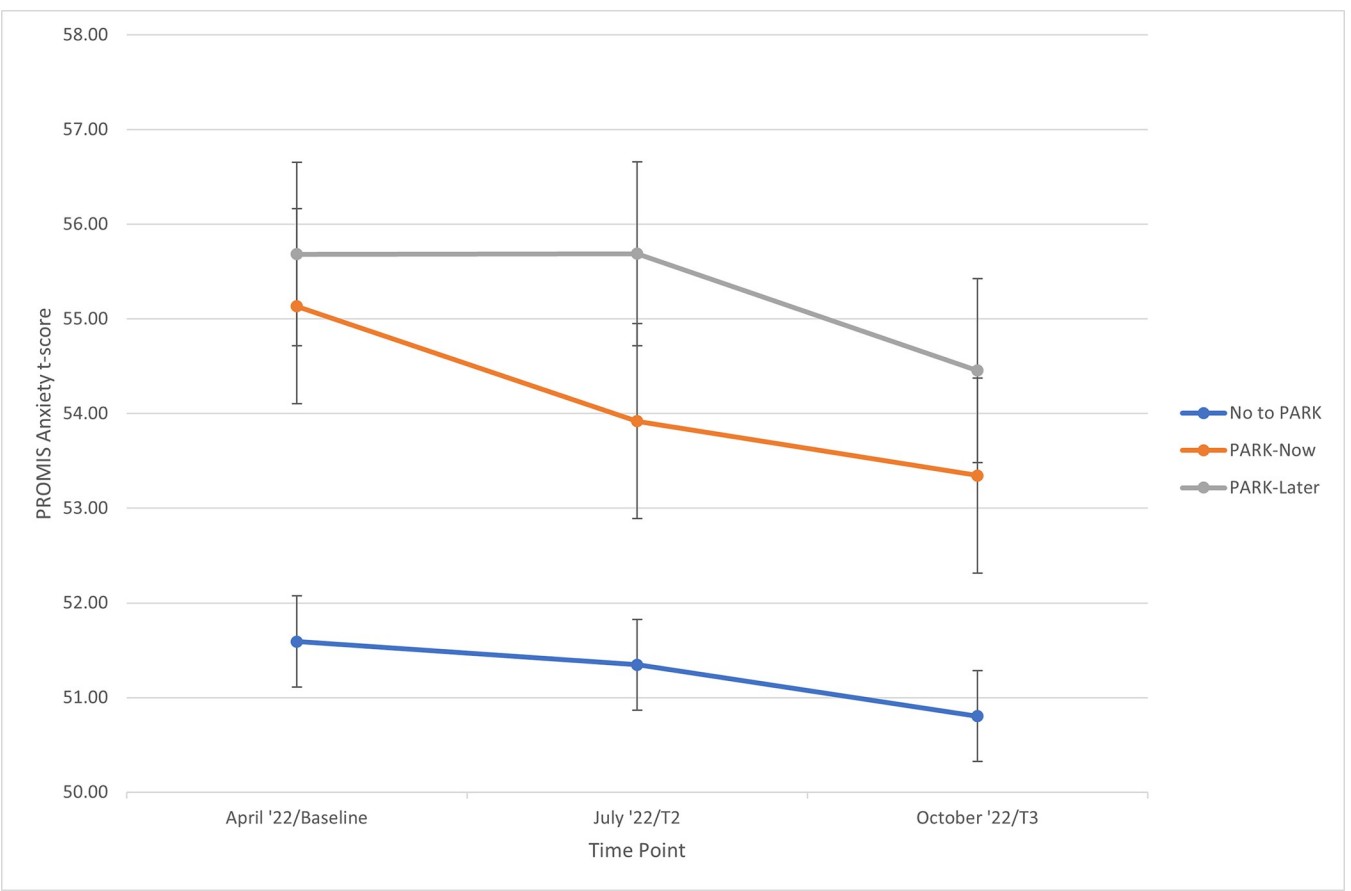

**Fig 3. Sample means with 95% CI for PROMIS anxiety t-scores by group at each timepoint.**

after the waitlist period. In the full, combined PARK sample, once participants were given access to the platform, 9.4% (52 of 554) completed all 5 lessons, another 47.8% (n = 265), completed one to four of the weekly skills lessons, and 42.8% did not complete any of the lessons (n = 237). The usage groups did not differ on any baseline variables except age (p = .01); the group accessing all 5 skills was older compared to the other groups (See S1 Table).

Results for the As-Treated analyses with usage group as a moderator of intervention effect on change in wellbeing outcomes are shown in Table 4. Usage group was a significant predictor of change in positive affect from pre- to post-intervention (T1 to T2 for PARK-Now and T2 to T3 for PARK-Later) such that those who completed all five weekly skill lessons of the intervention improved more compared to the no usage group (0 lessons completed); specifically, participants who completed all weeks of the intervention improved 2.95 points on positive affect, while those who did not use the intervention did not change from pre- to post-intervention (p = 0.001). The effect size for this difference in change is in the medium range (Cohen's d = .50) and is approaching a meaningful change for PROMIS scores [50]. Differences by usage group are included in S1 Fig. Although within-group improvements for the group that completed all lessons was also significant for meaning and purpose (p = .05) and sleep (p = .01), these changes were not significantly better than in the no usage group. There was also a significant within-group improvement for depression (p = .017) in the no usage group, although this did not differ from the other usage groups either. There were no statistically significant usage group differences in change from pre- to post-intervention for any other outcome.

**Table 4. As-treated analyses with PARK-Now and PARK-Later combined.**

| Sessions Completed | | Pre | | Post | | Change | | | P-value for Change | p-value for interaction | Cohen's d effect size |
|---|---|---|---|---|---|---|---|---|---|---|---|
| | | Mean | Std | Mean | Std | Mean | Std | 95% CI | | | |
| **PROMIS—Positive Affect** | **0** | 47.16 | 0.58 | 47.26 | 0.58 | 0.10 | 0.37 | (-0.63,0.82) | 0.796 | reference | |
| | **1–4** | 46.15 | 0.55 | 46.31 | 0.55 | 0.16 | 0.35 | (-0.52,0.85) | 0.646 | 0.90 | 0.01 |
| | **All 5** | 45.54 | 1.23 | 48.49 | 1.23 | 2.95 | 0.79 | (1.40,4.50) | 0.0002 | 0.001 | 0.50 |
| **PROMIS—Meaning and Purpose** | **0** | 52.12 | 0.68 | 52.27 | 0.68 | 0.15 | 0.35 | (-0.53,0.83) | 0.663 | reference | |
| | **1–4** | 51.78 | 0.65 | 51.86 | 0.65 | 0.08 | 0.33 | (-0.56,0.72) | 0.800 | 0.89 | -0.01 |
| | **All 5** | 49.88 | 1.46 | 51.32 | 1.46 | 1.45 | 0.74 | (0.00,2.90) | 0.050 | 0.11 | 0.24 |
| **PROMIS—Depression** | **0** | 50.57 | 0.47 | 49.79 | 0.47 | -0.77 | 0.32 | (-1.41,-0.14) | 0.017 | reference | |
| | **1–4** | 51.05 | 0.45 | 50.65 | 0.45 | -0.40 | 0.31 | (-1.00,0.20) | 0.192 | 0.40 | 0.07 |
| | **All 5** | 52.09 | 1.01 | 51.41 | 1.01 | -0.68 | 0.69 | (-2.04,0.68) | 0.324 | 0.91 | 0.02 |
| **PROMIS—Anxiety** | **0** | 55.36 | 0.50 | 54.49 | 0.50 | -0.88 | 0.29 | (-1.45,-0.30) | 0.003 | reference | |
| | **1–4** | 55.33 | 0.47 | 54.89 | 0.47 | -0.44 | 0.28 | (-0.98,0.10) | 0.113 | 0.28 | 0.10 |
| | **All 5** | 56.12 | 1.06 | 55.19 | 1.06 | -0.93 | 0.62 | (-2.16,0.29) | 0.135 | 0.93 | -0.01 |
| **PROMIS—Social Roles** | **0** | 45.89 | 0.59 | 45.75 | 0.59 | -0.15 | 0.36 | (-0.86,0.57) | 0.686 | reference | |
| | **1–4** | 46.01 | 0.56 | 45.70 | 0.56 | -0.31 | 0.34 | (-0.98,0.37) | 0.369 | 0.75 | -0.03 |
| | **All 5** | 46.43 | 1.26 | 45.69 | 1.26 | -0.74 | 0.78 | (-2.26,0.79) | 0.342 | 0.49 | -0.11 |
| **Burnout—Total Score** | **0** | 37.80 | 0.50 | 37.72 | 0.50 | -0.08 | 0.26 | (-0.59,0.42) | 0.743 | reference | |
| | **1–4** | 37.07 | 0.47 | 36.80 | 0.47 | -0.27 | 0.24 | (-0.74,0.21) | 0.268 | 0.60 | -0.05 |
| | **All 5** | 37.27 | 1.07 | 36.24 | 1.07 | -1.03 | 0.55 | (-2.11,0.04) | 0.059 | 0.12 | -0.24 |
| **PROMIS—Sleep** | **0** | 57.44 | 0.48 | 57.81 | 0.48 | 0.36 | 0.27 | (-0.17,0.90) | 0.182 | reference | |
| | **1–4** | 57.70 | 0.45 | 58.30 | 0.45 | 0.60 | 0.26 | (0.10,1.11) | 0.019 | 0.52 | 0.06 |
| | **All 5** | 58.59 | 1.02 | 60.08 | 1.02 | 1.50 | 0.58 | (0.36,2.64) | 0.010 | 0.08 | 0.27 |

Note: 0 = zero of 5 online skill lessons completed; 1–4 = one to four lessons completed, all 5 = all five lessons completed

Means provided are least-squares means estimated from the models

**Moderators in intent to treat analyses.** In addition to usage, we examined a number of other variables as potential moderators of intervention effect on positive affect to see if there were some groups for whom PARK was more effective. Age, gender, and HCW role did not moderate the effect of PARK on positive affect. In contrast, baseline meaning and purpose, depression, anxiety, social isolation, and burnout were all significant predictors of change in positive affect over time (See S2 Table), and thus were tested individually as moderators in the ITT analyses with positive affect as the outcome. Each of these factors also significantly moderated the effect of the PARK intervention on change in positive affect over the study period, such that those with "worse" emotional wellbeing (lower meaning and purpose, higher depression, anxiety, social isolation, or burnout) at baseline had more significant improvement in positive affect in the intervention group compared to the PARK-later controls (See S3 Table).

**Acceptability and feasibility.** A subset (n = 127) of PARK participants responded to follow-up questions about reasons for not completing lessons or skills practice (See S4 Table) and these responses provided important feedback regarding the acceptability of PARK in its current untailored form for HCW. Over half (52.8%) of participants reported that they did not have enough time to complete all the skill lessons; 29% said they forgot, and 30% reported that they lost interest (participants could select multiple reasons for not completing the skills). The most frequently cited reasons for not completing the skills practice included not enough time (50.4%), forgetting to do it (34.6%), losing interest (21.3%), and finding the daily practice too demanding (23.6%).

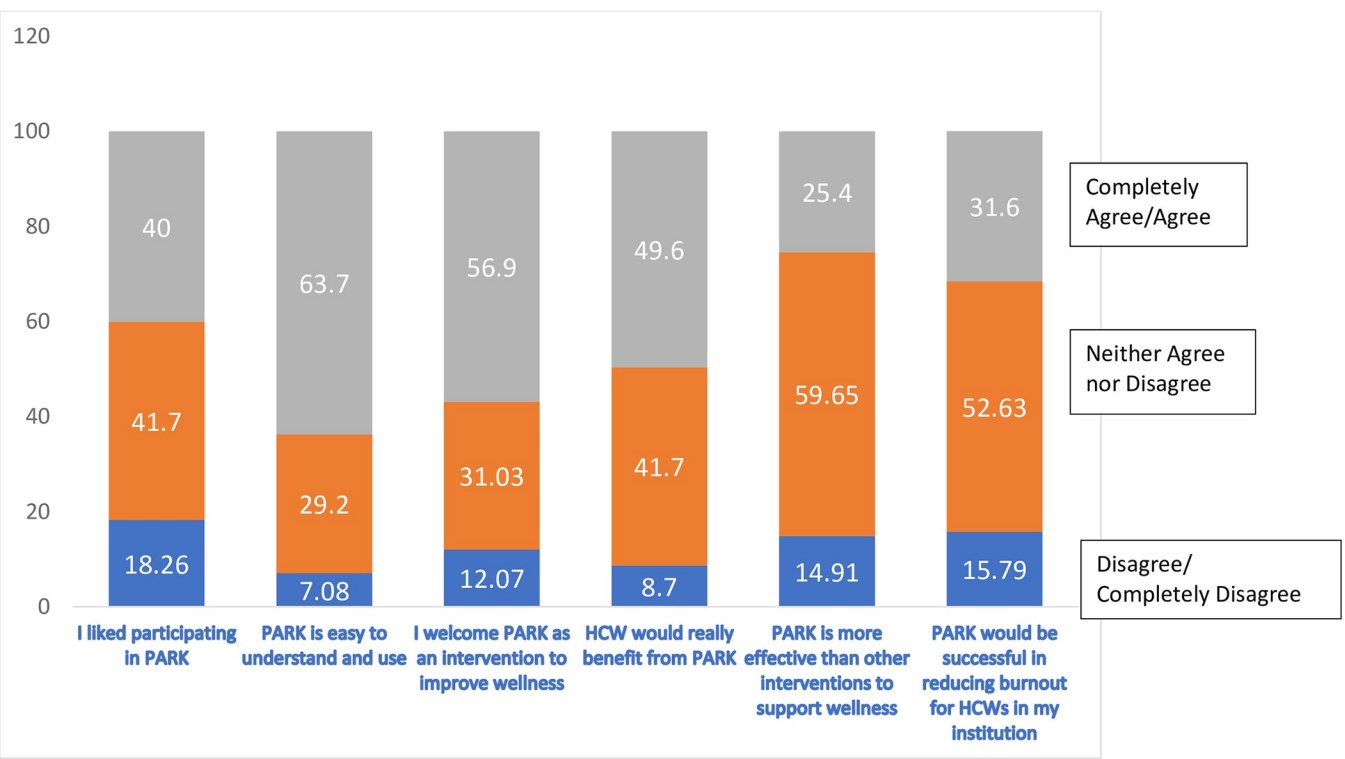

**Fig 4. Acceptability ratings for PARK online platform (N = 115).**

Ratings on the acceptability questions are in Fig 4 and indicated that participants generally held favorable or neutral opinions about aspects of PARK for HCW. Participants generally agreed that *PARK is easy to understand and use* (63.7% completely agree or agree) and that they would *welcome PARK as an intervention to improve wellness* (56.9% completely agree or agree). Almost half (49.6%) agreed or completely agreed with the statement *HCW would really benefit from PARK* but an additional 41.7% neither agreed nor disagreed with that statement. Whereas 40% said agree or completely agree with the statement *I liked participating in PARK*, another 41.7% said that they neither agreed nor disagreed with this statement and just over 18% disagreed or completely disagreed. Participants were also noncommittal as a group on the statements *PARK is more effective than other interventions to support wellness* (59.65% neither agreed nor disagreed) and *PARK would be successful in reducing burnout for HCWs in my institution* (52.63% neither agreed nor disagreed).

Feedback on the open-ended question about how we could improve PARK for HCW noted some logistical issues with the program like participants not receiving the initial email to login to the platform or not recalling enrolling in PARK. *"I totally forgot that I opted in and ignored the emails because of my busy schedules. I wish I made more time to participate as I intended. I would have loved to see if it worked."*

Others found the daily reminders to practice the skills too demanding. *"My intention was to fully participate in this program, but I didn't realize how demanding daily entries would be. I missed a few and then it seemed impossible to catch-up and I lost interest, but would love to have an opportunity to participate again now that I understand the time requirement."*

Finally, feedback reflected a common concern for the mismatch between individually-directed interventions like PARK used to address the systems-level factors that drive burnout.

*"I did not participate at all due to low staffing and no time to complete a program like this. I think these interventions are very important but if the organization does not specifically prioritize and support those who need it most, blocking out specific time and providing staffing, it will not be successful."*

*"..the reason 'wellness' is so chronically low among healthcare workers isn't because we're not doing enough mindfulness modules online. Its because we are expected to work too much, we put our lives on the line during a pandemic. . .. An online module was just an extra thing for me to do on top of the billions of other tasks I had to do. . .*

## Discussion

COVID-19 further exacerbated burnout that was already on the rise prior to the onset of the pandemic in 2020 [56] and put additional stress on HCWs who were working within systems that were not designed to support them through an unprecedented global health crisis. Given the urgent need for coping resources for HCW and evidence that interventions that target positive affect can promote resilience [39, 40–43], we offered PARK, a self-guided online positive psychological intervention, to a cohort of HCW to help buffer the impact of stress and burnout on their psychological wellbeing. Results of our randomized waitlist-controlled trial of PARK lead to several conclusions. First, those who chose to enroll in PARK had significantly poorer wellbeing (i.e., lower positive affect and meaning and purpose, higher levels of burnout, anxiety, depression, and social isolation) compared to those who did not enroll, indicating there is a self-identified demand for programs like PARK, especially among those who may be struggling more to cope well with stress. Those who had poorer wellbeing at baseline—lower meaning and purpose, higher depression, anxiety, social isolation, and more burnout—had greater improvements in positive affect compared to those who started out with higher levels of wellbeing. Second, although the intent to treat analyses did not demonstrate a statistically significant impact of PARK in the intervention compared to the waitlist control, follow-up as-treated analysis demonstrated that those who received a larger "dose" of the intervention (completing all 5 weeks of skills compared to zero) reported statistically significant improvements in our primary outcome of positive affect. Barriers to participation included lack of time and forgetting to access the skill lessons and/or practice exercises. Participants made a number of suggestions to improve the implementation and effectiveness of PARK, particularly in concert with systems-level interventions to better support HCW.

The COVID-19 pandemic prompted a proliferation of interventions that targeted individual behaviors to help HCW increase resilience to stress and burnout. Tests of these various interventions produced mixed findings, likely due to the wide variation in content, length, and delivery mode [57–59]. Results from the current trial indicate that PARK may have been effective for some of the participants–namely for those with higher burnout and poorer wellbeing. In both the present PARK HCW trial and our previous trial of PARK in the general population [39], participants who completed more skills improved more on measures of wellbeing. In the general population sample, those who accessed at least one skill had significantly greater decreases in anxiety and social isolation compared to those who did not complete any skills. In the HCW cohort, those who completed all the skills (9% of the sample) had significantly greater improvements in positive affect compared to those who did not complete any skills.

One third (554/1685) of the HCW cohort who responded to the April 2022 survey enrolled in PARK, but only 57% of those (317/554) completed at least 1 lesson. Rates of participation in interventions that aim to combat stress and burnout among HCW are generally low, even

when the interventions are intentionally kept brief [60] and these challenges aren't unique to samples of HCWs. Retention in digitally-delivered interventions more generally is as low as 50% in randomized controlled trials and significantly lower for open access apps outside the research context [61, 62]. A challenge to future work is to reach and retain the significant number of individuals who may enroll in a program such as PARK and develop strategies to maintain interest in the intervention and reduce barriers for full participation.

A subset of the participants who enrolled in PARK provided feedback and suggestions for improving the user experience including increasing portability of the program by linking it to personal rather than work email; providing PARK as an app that sends notifications to engage with the program vs an email prompt; and reducing the demands of the intervention. In addition to these suggestions, offering protected time for HCW to participate during the workday and options to connect as a group to discuss the skills helped increase engagement in other samples of HCW [63]. Finally, expansion of outcome measures to include engagement, broader categories of quality of life, and fatigue has been suggested [64] and may better capture the impact of PARK in HCW.

The present study had a number of strengths. It was a randomized controlled trial with a relatively large sample size that offered an evidence-based option to help HCW cope with the ongoing stress of the COVID-19 pandemic. We measured a number of outcomes using PROMIS measures which allows for comparison to general population norms. In contrast to previous studies of stress-reduction interventions among HCW that restricted samples to only nurses or physicians [65–70], we enrolled healthcare workers from a wider variety of roles and demonstrated that the impact of the intervention did not differ based on role. Resilience and burnout prevention are important for all those who work in the healthcare system. An additional strength was the use of a multi-component intervention; PARK includes eight different skills so individuals can find the ones that work best for them. A majority of interventions that have been tested for prevention of HCW burnout focused solely on mindfulness [67, 68]. Whereas PARK has a mindfulness component, it also includes several other positive emotion regulation skills, and thus offers options for those who may not be particularly drawn to mindfulness or who otherwise may find it difficult to tolerate.

Despite these strengths, this present study has a number of limitations including, a lack of racial and ethnic diversity in the sample and recruitment from within an ongoing cohort study which limits the conclusions that can be drawn regarding the generalizability to the broader population of HCWs. Furthermore, the follow-up time was not sufficient to assess any long-term impact of the intervention on wellbeing. Previous tests of this positive emotion intervention in other samples have shown that the improvements in wellbeing in the intervention compared to control conditions grew with longer follow-up [41], suggesting that additional time to practice the skills and incorporate them into a daily routine, may prove beneficial. Finally, the timing of the intervention with respect to the ongoing COVID-19 pandemic likely was not optimal as the HCW were already overburdened by work demands. Despite the need for some relief from the stress, 2 years into an ongoing crisis in the healthcare system prompted by COVID, was not ideal timing for implementation. Future studies could examine the impact of PARK incorporated during onboarding (e.g. [71]) such that the "toolbox" of coping skills could already be established when the need becomes acute as it did with the pandemic.

Finally, and critically, while PARK offers one individual-level solution that has the potential to address HCW stress levels, the qualitative feedback suggested that PARK is more likely to be successful at reducing burnout if implemented in parallel with changes to system-level policies. As one participant noted "we need basic needs met (adequate staffing and work expectations and time for lunch and bathroom breaks) prior to most staff being able to participate in intervention such as this." It is clear that Individual-level solutions are unlikely to be powerful

enough to completely counteract the significant, structural problems within the U.S. health-care system. Instead, individual-level interventions like PARK should be couched within larger systems-level changes that target team- or clinic-level programs such as improving efficiencies or increased staffing in workflow in order to maximize the potential to improve job satisfaction and reduce burnout in healthcare workers [72, 73].

## Conclusions

We implemented PARK, a self-guided online positive psychological intervention, in a cohort of HCW to help buffer the impact of stress and burnout on their psychological wellbeing. Results of our randomized waitlist-controlled trial indicate that PARK can be effective in those with a high mental health need and future work will focus on adaptations to increase engagement and tailor PARK for HCWs who could most benefit.

## Supporting information

**S1 Checklist. CONSORT 2010 checklist of information to include when reporting a randomised trial\*.**
(DOC)

**S1 Table. Pre-intervention comparisons of demographics, burnout, and well being variables by usage group.**
(DOCX)

**S2 Table. Baseline moderators of change in positive affect.**
(DOCX)

**S3 Table. Intent-to-treat analyses testing including baseline moderators of PARK effects on positive affect.**
(DOCX)

**S4 Table. Reasons for not completing skill lessons or practice (N = 127).**
(DOCX)

**S1 Fig. Positive affect pre-and post-PARK by usage group.**
(DOCX)

**S1 File.**
(PDF)

## Author Contributions

**Conceptualization:** Judith Tedlie Moskowitz, Elizabeth L. Addington, Amisha Wallia, Lisa R. Hirschhorn, John T. Wilkins, Charlesnika Evans.

**Data curation:** Peter Cummings, Melanie E. Freedman, Cerina Lee, Thanh Huyen Vu.

**Formal analysis:** Kathryn L. Jackson.

**Funding acquisition:** Judith Tedlie Moskowitz, Amisha Wallia, Lisa R. Hirschhorn, John T. Wilkins, Charlesnika Evans.

**Investigation:** Melanie E. Freedman, Jacquelyn Bannon, Cerina Lee, Thanh Huyen Vu.

**Methodology:** Judith Tedlie Moskowitz, Elizabeth L. Addington, Amisha Wallia, Lisa R. Hirschhorn, John T. Wilkins, Charlesnika Evans.

**Project administration:** Melanie E. Freedman, Cerina Lee.

**Resources:** Judith Tedlie Moskowitz.

**Supervision:** Judith Tedlie Moskowitz, Amisha Wallia, Lisa R. Hirschhorn, John T. Wilkins.

**Validation:** Kathryn L. Jackson, Peter Cummings, Thanh Huyen Vu, Charlesnika Evans.

**Visualization:** Peter Cummings.

**Writing – original draft:** Judith Tedlie Moskowitz, Kathryn L. Jackson, Peter Cummings, Elizabeth L. Addington, Melanie E. Freedman.

**Writing – review & editing:** Judith Tedlie Moskowitz, Kathryn L. Jackson, Peter Cummings, Elizabeth L. Addington, Melanie E. Freedman, Jacquelyn Bannon, Cerina Lee, Thanh Huyen Vu, Amisha Wallia, Lisa R. Hirschhorn, John T. Wilkins, Charlesnika Evans.

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
