## [Decision Letter · Decision Letter 0]

13 Nov 2023

PONE-D-23-29506Feasibility, acceptability, and efficacy of a positive emotion regulation intervention to promote resilience for healthcare workers during the COVID-19 pandemic: A randomized controlled trialPLOS ONE

Dear Dr. Moskowitz,

Thank you for submitting your manuscript to PLOS ONE. After careful consideration, we feel that it has merit but does not fully meet PLOS ONE’s publication criteria as it currently stands. Therefore, we invite you to submit a revised version of the manuscript that addresses the points raised during the review process.

We look forward to receiving your revised manuscript.

Kind regards,

Mohammed Amir Rais, DMD

Academic Editor

PLOS ONE

Additional Editor Comments:

Hi!

Please read carefully reviewers remarks and correct your paper in 10 days and send it back.

Thanks

Reviewers' comments:

Reviewer's Responses to Questions

**Comments to the Author**

1. Is the manuscript technically sound, and do the data support the conclusions?

Reviewer #1: Yes

Reviewer #2: Yes

Reviewer #3: Yes

2. Has the statistical analysis been performed appropriately and rigorously? 

Reviewer #1: Yes

Reviewer #2: Yes

Reviewer #3: Yes

3. Have the authors made all data underlying the findings in their manuscript fully available?

Reviewer #1: Yes

Reviewer #2: Yes

Reviewer #3: No

4. Is the manuscript presented in an intelligible fashion and written in standard English?

Reviewer #1: No

Reviewer #2: Yes

Reviewer #3: Yes

5. Review Comments to the Author

Reviewer #1: This manuscript is written well and has good corelation. But I recommend to add some valid points before acceptance of this paper.

Add the characteristis of participants in tabulated form.

Add the conclusion, after discussion.

Also write the future prospectives of this study.

Reviewer #2: In general, this research work is quite strong and could provide robust evidence to this area of research

Abstract:

In the background section of abstract, authors need to include a “Specific” objective for the trial.

In the methods section of the abstract, authors should provide information on the number of participants, setting and few details on the intervention and statistical methods used in analysis.

Introduction:

This section is very well-written.

Methods:

In line 193, the authors referred to time using two labels (T1 and T2). It was not added explicitly what does each of the two Ts denote?

In line 206, what did authors mean by “Combined sample”? Also, there are some words repeated in those few lines to be rephrased.

Results and Discussion sections are very well written

However, in the CONSORT diagram, I am a bit confused how did the number of HCWs increased in PARK NOW gp. from 122 at T2 to 184 at T3?

Also in Fig. 2a it was clear that the primary outcome has fallen in PARK NOW group between T2 and T3. Authors need to elaborate in that in their discussion and recommendation sections.

In Table 3 and 4, authors need to add confidence intervals for changes occurring in different variables in study gps.

I recommend Authors would add at least two clear conclusions based on the research findings.

Reviewer #3: Add statistics to the abstract in addition to the significance /non-significance.

Better name the model as “ linear mixed-effects model” or “linear mixed model”. Be consistent with the model name throughout the text.

You can not “independently” add a moderator to the model.

T score is not clearly explained.

Since LSM is the estimate for primary analysis, better report it in the tables.

The flow chart is not clear to me. Omit usage weeks. Add reasons for number changes and add numbers for analysis. See examples in the journal.

It is hard to see which groups have more changes in figures 2b and 3b which can be omitted.

6. PLOS authors have the option to publish the peer review history of their article (what does this mean?). If published, this will include your full peer review and any attached files.

Reviewer #1: **Yes: **Kanwal Irshad

Reviewer #2: No

Reviewer #3: No

---

## [Author Response · Author response to Decision Letter 0]

22 Dec 2023

Response to reviewers and editors are included in the response to reviewers document and also pasted here:

We appreciate the editor and reviewers’ comments and have revised the manuscript accordingly. We address each comment below. Thank you for your time and attention.

Reviewer #1: This manuscript is written well and has good corelation. But I recommend to add some valid points before acceptance of this paper.

• Add the characteristis of participants in tabulated form. 

See Table 2.

• Add the conclusion, after discussion.

We have added a conclusion section to the Discussion. 

• Also write the future prospectives of this study.

We have added a sentence about future work to the conclusion section. 

Reviewer #2: In general, this research work is quite strong and could provide robust evidence to this area of research

Abstract:

• In the background section of abstract, authors need to include a “Specific” objective for the trial.

We have added an objective sentence to the abstract.

• In the methods section of the abstract, authors should provide information on the number of participants, setting and few details on the intervention and statistical methods used in analysis.

Added to the abstract.

Introduction:

• This section is very well-written.

Methods:

• In line 193, the authors referred to time using two labels (T1 and T2). It was not added explicitly what does each of the two Ts denote? 

Earlier in the manuscript we describe what T1, T2, and T3 denote as follows: Baseline/T1 completed between April 1, 2022 and April 30, 2022, July 2022 (T2), and October 2022 (T3). We have reiterated this in the analysis section as requested by the reviewer.

• In line 206, what did authors mean by “Combined sample”? Also, there are some words repeated in those few lines to be rephrased. 

We revised this section to clarify that the “combined sample” was all participants, whether randomized to PARK-Now or PARK-Later. This full sample of all participants was analyzed together to example change from the assessment just prior to starting the intervention to the assessment following the intervention (T1 to T2 for PARK-Now and T2 to T3 for PARK-Later).

Results and Discussion sections are very well written

• However, in the CONSORT diagram, I am a bit confused how did the number of HCWs increased in PARK NOW gp. from 122 at T2 to 184 at T3? 

These numbers reflect the number of participants who completed the survey at each time point. Even if they did not complete the survey at T2, they were still invited to complete the survey at T3. Thus, a number of participants who skipped the T2 survey, completed the T3 survey leading the N to increase from T2 to T3. 

• Also in Fig. 2a it was clear that the primary outcome has fallen in PARK NOW group between T2 and T3. Authors need to elaborate in that in their discussion and recommendation sections. 

• While both PARKnow and PARKlater decrease from T2 to T3, the decrease is not statistically different from 0, and the difference in decrease between the two groups is not statistically different, either. We have added this point to the results. 

• In Table 3 and 4, authors need to add confidence intervals for changes occurring in different variables in study gps.

95% Confidence Intervals have been added to Table 3 and Table 4.

• I recommend Authors would add at least two clear conclusions based on the research findings.

We have added a conclusion section to the end of the Discussion.

Reviewer #3 

• Add statistics to the abstract in addition to the significance /non-significance. Better name the model as “ linear mixed-effects model” or “linear mixed model”. Be consistent with the model name throughout the text.

Test statistics and p-values have been added to the abstract, and language has been changed such that the term “mixed-effects linear regression” is now used throughout.

• You can not “independently” add a moderator to the model

This has been updated to “individually”, and specific clarification has been added to indicate that one model per moderator was run for this analysis. 

• T score is not clearly explained. 

We have added the following clarification: “Moderators originally scored on a T-score metric (i.e., PROMIS measures) were converted to theta scores (theta=(t-score-50)/10)) to provide a meaningful "zero” for effects.”

• Since LSM is the estimate for primary analysis, better report it in the tables. 

LSM estimates are presented in all tables. Footnotes have been added to each table for additional clarification (e.g., “Note: Means provided are least-squares means estimated from the models”)

• The flow chart is not clear to me. Omit usage weeks. Add reasons for number changes 

and add numbers for analysis. See examples in the journal. 

We have added to the description of Figure 1 (the flow chart) in the text. Figure 1 is the CONSORT diagram following the guidance published in 2010: Schulz KF, Altman DG, Moher D, for the CONSORT Group (2010) CONSORT 2010 Statement: Updated Guidelines for Reporting Parallel Group Randomised Trials. PLoS Med 7(3): e1000251. The usage weeks are akin to the amount of the intervention “dose” received. The numbers in the diagram correspond to the Ns analyzed at each time point. 

It is hard to see which groups have more changes in figures 2b and 3b which can be omitted. We have omitted the waterfall plots in the main text (Figures 2b and 3b) as well as the supplementary data section. 

We have double-checked to make sure the manuscript follows PLOS ONE’s style requirements.

We have uploaded the de-identified dataset to OSF: https://osf.io/bvdu5/that and it will be made publicly available once the manuscript is accepted for publication. 

We have added information on the name of the IRB, the approval number, and the consent process to the Methods section.

We have added a caption for each figure.

5. Please include captions for your Supporting Information files at the end of your manuscript, and update any in-text citations to match accordingly. 

We have updated the in-text citations and captions for supplemental figures and tables.

6. Please review your reference list to ensure that it is complete and correct 

We have checked the reference list and no changes are needed.

---

## [Decision Letter · Decision Letter 1]

27 May 2024

Feasibility, acceptability, and efficacy of a positive emotion regulation intervention to promote resilience for healthcare workers during the COVID-19 pandemic: A randomized controlled trial

PONE-D-23-29506R1

Dear Dr. Moskowitz

We’re pleased to inform you that your manuscript has been judged scientifically suitable for publication and will be formally accepted for publication once it meets all outstanding technical requirements. We appreciate your hard work and great dedication that you provided during revision to make this paper of high quality. Again, congratulations on your hard work, and looking forward to receiving other projects in future.

Kind regards,

Mohammed Amir Rais, DMD

Academic Editor

PLOS ONE

---

## [Editor Report · Acceptance letter]

13 Jun 2024

PONE-D-23-29506R1 

PLOS ONE

Dear Dr. Moskowitz, 

I'm pleased to inform you that your manuscript has been deemed suitable for publication in PLOS ONE. Congratulations! Your manuscript is now being handed over to our production team.

Kind regards, 

on behalf of

Dr. Mohammed Amir Rais 

Academic Editor

PLOS ONE